# On-demand transposition across light-matter interaction regimes in bosonic cQED

**Fernando Valadares** [1] ✉, **Ni-Ni Huang** [1], **Kyle Timothy Ng Chu**[1,2], **Aleksandr Dorogov** [1], **Weipin Chua**[3], **Lingda Kong**[1], **Pengtao Song** [1] & **Yvonne Y. Gao** [1,3] ✉

The diverse applications of light-matter interactions in science and technology stem from the qualitatively distinct ways these interactions manifest, prompting the development of physical platforms that can interchange between regimes on demand. Bosonic cQED employs the light field of high-Q superconducting cavities coupled to nonlinear circuit elements, harnessing the rich dynamics of their interaction for quantum information processing. However, implementing fast switching of the interaction regime without deteriorating the cavity coherence is a significant challenge. We present an experiment that achieves this feat, combining nanosecond-scale frequency tunability of a transmon coupled to a cavity with lifetime of hundreds of microseconds. Our implementation affords a range of useful capabilities for quantum information processing; from fast creation of cavity Fock states using resonant interaction and interchanging tomography techniques at qualitatively distinct interaction regimes on the fly, to the suppression of unwanted cavity-transmon dynamics during idle evolution. By bringing flux tunability into the bosonic cQED toolkit, our work opens up the possibility to probe the full range of light-matter interaction dynamics within a single platform and provides valuable pathways towards robust and versatile quantum information processing.

The interaction of light and matter is at the origin of numerous phenomena, from photosynthesis to photovoltaics, from the fundamental structures of atoms to quantum information processing. This ubiquitous physical concept is critical to understand nature and develop technologies. The coupled dynamics of light and matter span several qualitatively distinct regimes, from a direct swap of energy when both systems are resonant to a full decoupling when their frequencies differ significantly (Fig. 1 and Supplementary Fig. 1). Each regime has unique properties and advantages. A platform capable of harnessing all of this rich physics - and interchanging between regimes on demand - will grant powerful insights into both fundamental science and innovative applications.

Many physical platforms originate from the interactions between light and matter[1–5]. In particular, circuit quantum electrodynamics (cQED) stands out for its high engineerability and versatility[6]. In this system, the light field is built from a superconducting cavity, while nonlinear circuits based on Josephson junctions emulate the discrete energy spectrum of an atom[7]. Owing to the rapid advances in cQED hardware, it is now possible to build long-lived cavities and harness their rich dynamics through the manipulation of nonlinear elements such as transmons. This type of bosonic cQED architectures employs these continuous variable modes for quantum information processing[8,9]. It provides an ideal playground for exploring the qualitatively distinct behaviours of light-matter interactions, and a valuable platform for quantum error correction[10–12], analogue simulations[13,14] and metrology[15], and even for the long-term goal of engineering a universal quantum computer[9].

[1]Centre for Quantum Technologies, National University of Singapore, Singapore, Singapore. [2]Horizon Quantum Computing, Singapore, Singapore. [3]Department of Physics, National University of Singapore, Singapore, Singapore. ✉e-mail: fernando.valadares@u.nus.edu; yvonne.gao@nus.edu.sg

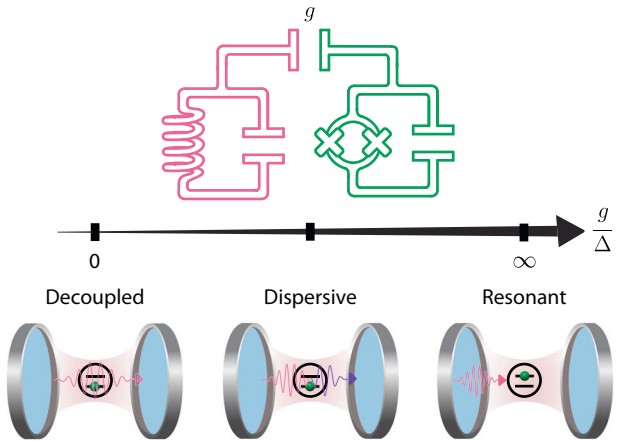

**Fig. 1 | Distinct regimes of light-matter interactions.** The dynamics of light and matter depend on their frequency detuning $\Delta$ and interaction strength $g$[58,59]. When $g/\Delta \to 0$, the systems are decoupled. When $g/\Delta \to \infty$, they are in the resonant regime and directly exchange energy (vacuum Rabi oscillations) at a rate $2g$. In between these limits is the dispersive regime, in which the energy of one system is shifted by excitations in the other. The dispersive interaction is considered strong (weak) when this shift is large (small) relative to the decoherence rates of the system[59]. We realize this model in bosonic cQED with a long-lived cavity acting as the light field (pink) coupled to a transmon with tunable frequency (green) representing matter.

While flux-controlled nonlinear coupling elements between cavities have been demonstrated recently[16,17], harnessing the full range of light-matter interaction regimes on demand remains an outstanding challenge in bosonic cQED. Although transmons can readily be made flux-tunable with the use of a superconducting quantum interference device (SQUID)[7], the addition of a strong, broadband flux line necessary to change the interaction dynamics within the coherence times of the system can critically reduce the cavity lifetime[18,19]. Because of this limitation, the community has so far focused operations in a single regime, which is usually either the strong or weak dispersive regime. These two regimes have qualitatively different dynamics which may make one more suitable than the other for specific applications[11,20]. A device that combines real-time tunable transmons to high-Q cavities would therefore pave the way to optimisation of existing techniques and the exploration of future quantum information processing protocols[21–27].

In this paper, we demonstrate on-demand transposition across several distinct interaction regimes of a transmon and a long-lived cavity at nanosecond timescales. This is achieved by engineering a metamaterial structure that can efficiently route magnetic flux to a tunable transmon[28,29] while preserving cavity coherences two orders of magnitude above that of the transmon. We showcase these features by mixing distinct coupling regimes within experiments that address central questions in quantum information processing. We prepare Fock states in the cavity in a vacuum Rabi experiment via resonant coupling with the transmon. We then perform tomography on the prepared states using both Wigner and characteristic functions by tuning the cavity-transmon interaction to the strong and weak dispersive regimes, respectively. Finally, we demonstrate the suppression of undesired nonlinear distortions and the mitigation of cavity decoherence from transmon interference by decoupling the two elements. Our work significantly expands the functionalities of bosonic cQED architectures and provides the means for more sophisticated applications of light-matter interactions.

## Results

We achieve fast tunability of the transmon frequency while preserving high cavity coherence times by incorporating a carefully designed magnetic hose[28,29] in a standard bosonic cQED device (Fig. 2a). The

system Hamiltonian is given by

$$
\frac{\hat{\mathbf{H}}(t)}{\hbar} = \omega_c \hat{\mathbf{a}}^\dagger \hat{\mathbf{a}} + \omega_t(t) \hat{\mathbf{b}}^\dagger \hat{\mathbf{b}} \\
+ g\left(\hat{\mathbf{a}}^\dagger \hat{\mathbf{b}} + \hat{\mathbf{a}}\hat{\mathbf{b}}^\dagger\right) - \frac{\alpha}{2}\hat{\mathbf{b}}^\dagger \hat{\mathbf{b}}^\dagger \hat{\mathbf{b}}\hat{\mathbf{b}},
\tag{1}
$$

where $\hat{\mathbf{a}}$ ($\hat{\mathbf{b}}$) is the cavity (transmon) mode annihilation operator, $\omega_c$ and $\omega_t(t)$ are the angular frequencies of the bare cavity and transmon, $g$ is the capacitive coupling factor between the circuits and $\alpha$ is the transmon anharmonicity. The tunability manifests as a change in the transmon-cavity detuning, $\Delta(t) = \omega_c - \omega_t(t)$ in response to the magnetic field provided by the hose.

The hose is an 8 mm-long cylinder made of 11 alternating layers of mu-metal and superconducting aluminium, with respective thicknesses of 0.15 mm and 0.10 mm, assembled around a 1 mm-diameter mu-metal core (see Supplementary Note 1). The layers are designed to not close on themselves around the cylinder, keeping a 0.5 mm gap along the hose axis to avoid superconducting loops. Due to the magnetic properties of this structure, the hose transfers magnetic fields with high efficiency, allowing the use of smaller, lower-impedance coils as the source. The source is made by winding 21 turns of superconducting wire into a planar coil with a maximum diameter of 3.4 mm (see inset in Fig. 2a). We double the magnetic field of the coil by stacking two such layers together and then attach them to one end of the hose. With this configuration, we can deliver a magnetic field of 80.6 nT/mA to the area of 1000 $\mu m^2$ forming the SQUID loop of the transmon.

To balance the efficient transfer of the magnetic field with the preservation of coherence times, we optimise the design of the bosonic cQED package, the transmon circuit and the magnetic hose itself. The lossy mu-metal layers of the hose are cut short 5 mm away from the end closest to the chip to minimize leakage of the cavity and transmon energy. The transmon pads are made short to prevent its field from spreading out towards the hose. The hose is then placed as far as possible from the cavity to prevent its energy from leaking out. Since the hose must be aligned with the SQUID loop, the transmon is also further separated from the cavity, reducing their coupling. Their interaction is recovered by placing a superconducting strip between both circuits that allows their fields to travel along the chip (Fig. 2a), reaching a coupling factor of $g/2\pi = 6.65$ MHz. A similar strip is placed between the transmon and the readout resonator to isolate the latter from the hose. Due to these design considerations, the cavity achieves average lifetimes of 200 $\mu s$, comparable to standard non-tunable bosonic cQED architectures found in the literature.

The effective execution of these delicate design choices of the hose and the coil results in full flux tunability of the transmon frequency from a maximum of $\omega_t^{max}/2\pi = 6.409$ GHz down to values close to zero. The cavity frequency is at $\omega_c/2\pi = 5.740$ GHz, so the transmon can access the resonant regime and also reach the decoupled regime at detuning $\Delta \gg g$. The small dimensions of the coil allow fast response to the drive. By connecting it to the digital-to-analogue converter port of a field-programmable gate array (FPGA), we can send fast flux pulses to the coil and tune the transmon over several hundreds of MHz at nanosecond timescales.

Reliably switching between cavity-transmon interaction regimes requires precise control of the transmon frequency. However, the flux pulse generated by the room-temperature electronics is distorted by impedances present in the wiring, causing the transmon frequency to deviate from the target trajectory. These distortions mainly come from the inductance of the coil and the radio-frequency (RF) port of the bias tee, and can be reverted by inverse digital filtering refs. 30,31. The method starts with characterizing the distorted frequency response to a step flux pulse using a pi-scope experiment, consisting of repeated transmon

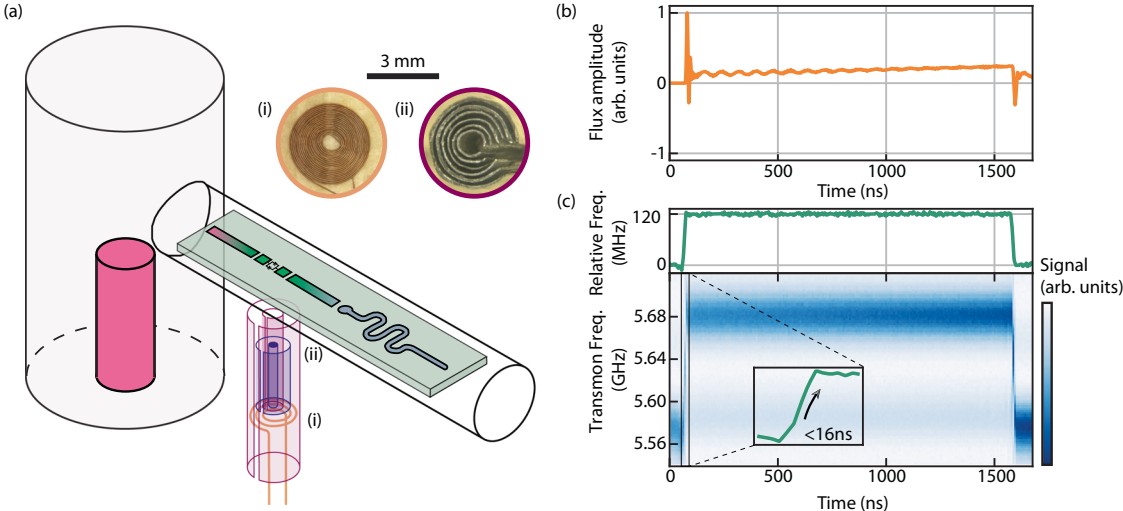

**Fig. 2 | Fast-flux incorporation to bosonic cQED. a** Our hardware consists of a coaxial stub and a transversal coaxline[60] hosting a SQUID-based transmon and a readout resonator. The structure is made with high-purity aluminium and mounted in a dilution refrigerator below 10 mK. The magnetic hose (not in proportion, see Supplementary Note 1) is composed of concentric aluminium (light purple) and mu-metal (blue) layers and is inserted perpendicularly to the coaxline, aligned with the SQUID loop. Insets show the microscope pictures of (i) the source coil and (ii) the hose end closest to the transmon. **b** Example of corrected step flux pulse generated at the FPGA. **c** Corresponding frequency response of the transmon. Inset resolves the rising edge, which is measured to be shorter than 16 ns.

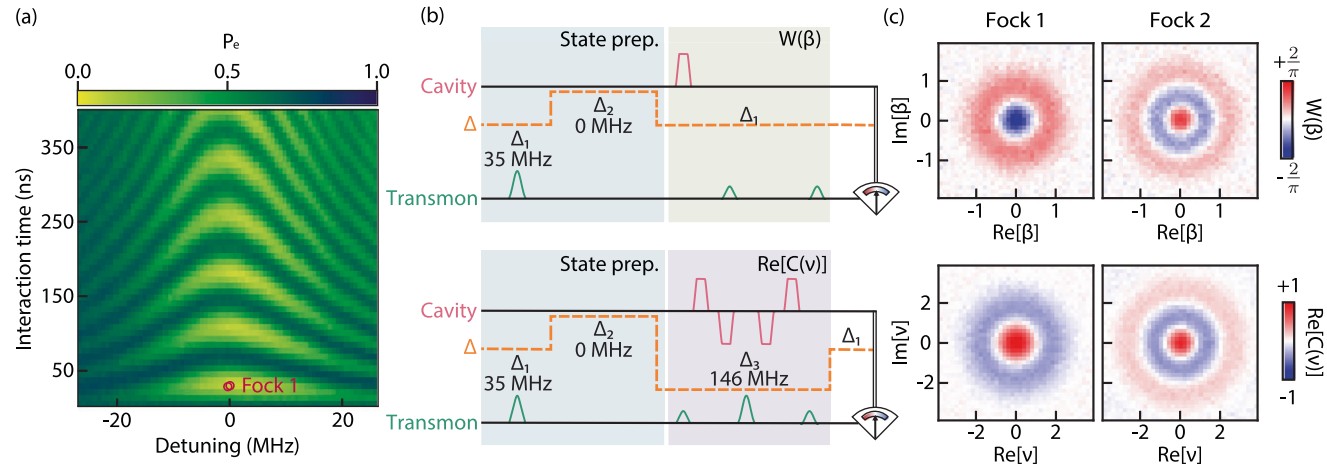

**Fig. 3 | Preparation and tomography of Fock states. a** Vacuum Rabi oscillations between cavity and transmon as a function of interaction time and detuning. Fock states $|1\rangle$ and $|2\rangle$ are prepared at resonance through direct exchange of excitation with the transmon. **b** Diagrams showing Fock state preparation and tomography protocols using dynamical tuning of the transmon. The transmon is prepared in $|e\rangle$ at $\Delta_1/2\pi = 35$ MHz, then put into resonance with the cavity. After the interaction, it is either returned to $\Delta_1$ for Wigner tomography $W(\beta)$ (top) or to $\Delta_3/2\pi = 146$ MHz to measure the real part of the characteristic function $C(\nu)$ (bottom). $\pi$ ($\frac{\pi}{2}$)-pulse is denoted by the large (small) transmon pulse. **c** Tomography of Fock states $|1\rangle$ and $|2\rangle$ by measuring the Wigner function (top) and real part of the characteristic function (bottom).

spectroscopies at different points of the trajectory. The current $I_c(t)$ at the coil is obtained by inverting the frequency response of a symmetric SQUID transmon $\omega_t(t) \approx (\omega_t^{max} + \alpha)\sqrt{|\cos(\pi k I_c/\Phi_0)|} - \alpha$[7], where $\Phi_0$ is the flux quantum and $\alpha/2\pi \approx 200$ MHz. Here, $k = 0.039$ $\Phi_0$/mA is a proportionality constant between the current and the magnetic flux threading the SQUID loop. From the $I_c(t)$ data, we digitally revert the distortions by training 11 first-order infinite-impulse response (IIR) and 2 finite-impulse response (FIR) filters. These filters are applied to a target flux pulse to produce the intended frequency trajectory. The corrected step pulse is shown in Fig. 2b, and the resulting frequency response reliably reproduces a step function as shown in Fig. 2c. The transmon frequency is stable over 1.5 $\mu$s up to a variation of $\pm 0.2$ %, and the rising edge has a duration below 16 ns. This switching time is much faster than the

timescale of $2\pi/g = 150$ ns, allowing nonadiabatic control of the interaction.

We use this fast and stable frequency control to demonstrate the creation of Fock states through vacuum Rabi oscillations (Fig. 3a). The transmon, initially detuned from the cavity, is excited from the ground state $|g\rangle$ to the next energy level $|e\rangle$. Then, a step flux pulse tunes the transmon nonadiabatically to the cavity frequency, at which point they start to coherently swap energy. After the interaction, the transmon is measured. If it is found in $|g\rangle$, the missing energy is assumed to be in the cavity, preparing Fock state $|1\rangle$. Figure 3a shows the chevron pattern of the vacuum Rabi experiment for variable interaction time and detuning. This result can only be obtained if the transmon tuning is fast compared to its coupling to the cavity; otherwise, the transmon state would transform adiabatically into joint eigenstates of the whole

system, which evolve trivially and do not show exchange of energy. Using this technique, state $|1\rangle$ is prepared in resonance after an interaction time $\tau_{int} \approx 30$ ns. Similarly, $|2\rangle$ is prepared by sequentially swapping two excitations from the transmon to the cavity. This process takes a total interaction time of $\tau_{int} \approx 50$ ns, as the Rabi frequency of the second transition is accelerated by a factor of $\sqrt{2}$[6]. This state preparation is much shorter than the relevant coherence times of the system and avoids the use of numerically-optimised pulses that are often necessary to create complex cavity states[32–34]. This experiment provides the basis for control techniques in the resonant regime[19,23] and more complex non-Gaussian states preparation, which are important resources for universal quantum computing[35].

To further showcase the versatility of our implementation, we characterize the Fock states with both Wigner and characteristic function tomographies by changing the transmon and cavity detuning. The ability to choose between these two protocols is an advantage since each gives ready access to different observables. In Wigner tomography, we directly probe the photon number parity of the bosonic state, a critical parameter in error correction protocols[36,37], and is typically optimal for systems in the strong dispersive regime. The characteristic function, on the other hand, measures the Fourier transform of the Wigner function and offers a more intuitive picture to study certain effects such as photon loss[38]. Contrary to Wigner, this tomography is usually done in the weak dispersive regime to decrease the nonlinear distortions during the protocol. The two techniques also have different resilience to errors. For example, the characteristic function protocol uses echoed conditional displacement (ECD) gates to cancel out the effects of low-frequency noise[34]. We measure the prepared $|1\rangle$ and $|2\rangle$ states using both tomographies by tuning the transmon to either strong or weak dispersive regimes (Fig. 3c). The results show the high quality of the state preparation and that our device can reliably perform tomography at different interaction regimes on demand. The ability to tune Hamiltonian parameters on the fly is thus a powerful tool for tailoring the tomography protocol to the requirements of any experiment.

Beyond the creation of non-Gaussian states and flexible tomography, another critical requirement of quantum information processing is the protection of the quantum state during idle evolution. In the dispersive regime ($g \ll \Delta$), the dynamics of the cavity are governed by $-\chi \hat{a}^\dagger \hat{a} \hat{b}^\dagger \hat{b} - \frac{K}{2} \hat{a}^\dagger \hat{a}^\dagger \hat{a} \hat{a}$, where $\chi$ is the dispersive shift between cavity and transmon and $K$ is the self-Kerr of the cavity. Although the self-Kerr is typically small, these extra dynamics can quickly distort the cavity state, motivating the search for ways to control and mitigate nonlinearities using drives or alternative circuits[16,39–42]. Using fast-flux tunability, we can suppress the self-Kerr effect in the bosonic state by simply decoupling it from the transmon (Fig. 4a). We demonstrate this suppression with the evolution of a coherent state $|\alpha\rangle$ with an average photon number of $|\alpha|^2 \approx 6.25$. We allow the state to evolve freely for 10 $\mu$s at two different flux points with $K/2\pi \approx 0.09$ kHz and $K/2\pi \approx 6$ kHz. Details for every flux point used are given in Supplementary Note 1. After this interval, we tune the transmon frequency to measure the imaginary part of the characteristic function. The tomography reveals a stark contrast between the cavity dynamics at each point as shown in Fig. 4a. At the high self-Kerr point, the state is strongly distorted, losing the well-defined phase of a coherent state. In comparison, the evolution at low self-Kerr keeps the coherent state virtually unchanged, with its properties preserved due to the suppressed cavity-transmon interactions.

The same strategy is used to tune down the dispersive interaction and mitigate the propagation of transmon errors to the bosonic state. We simulate a scenario where fluctuations in the transmon energy cause dephasing in the cavity (Fig. 4b). Here, the transmon is initialized in $|g\rangle$ and the cavity is prepared in $|\alpha\rangle$. We apply a pulse to put the transmon in a superposition $(|g\rangle + |e\rangle)/\sqrt{2}$ and allow the system to evolve for $\tau = 400$ ns. Due to the dispersive shift $\chi$ of the cavity

frequency when the transmon is excited, the system forms an entangled state $|g, \alpha\rangle + |e, \alpha e^{i\chi\tau}\rangle$ (up to a normalization factor), which continuously decoheres into a statistical mixture of $|g, \alpha\rangle$ and $|e, \alpha e^{i\chi\tau}\rangle$ due to transmon decoherence. This procedure is repeated 10 times, inducing a random walk of the phase of $\alpha$ with steps of $\chi\tau$. At the end of the procedure, the transmon is projected onto $|g\rangle$ and the real part of the characteristic function is measured. We compare the evolution of the cavity state for two different flux points. In the first case, the dispersive shift is strongly suppressed to $\chi/2\pi \approx 0.05$ MHz and the real characteristic function of the final state is shown in Fig. 4b (left panel). Due to the decoupling, the coherent state still preserves a well-defined phase given by the direction of the fringes, with an overall rotation due to the residual $\chi$. In the second case, the dynamics of the cavity are strongly intertwined with the transmon at $\chi/2\pi = 0.94$ MHz, resulting in a much stronger distortion of the state. As such, the phase of the cavity state is completely scrambled and the fringes of the coherent state are fully washed out (Fig. 4b).

## Discussion

These experiments show a high-Q bosonic cQED design that leverages the many regimes of light-matter interactions and combines them to create versatile functionalities. The resourcefulness of our solution is demonstrated in multiple cases of interest for quantum information processing. We perform vacuum Rabi oscillations by nonadiabatically switching the transmon and cavity into resonance and use this technique to prepare Fock states without the need for numerically optimised pulses. The states are measured with both Wigner and characteristic function, respectively in the strong and weak dispersive regimes, showing flexibility in extracting information from the cavity. We then decouple the cavity and transmon to show how to freeze the evolution of the cavity state and protect it against spurious transmon dynamics during idle times.

As a first proof-of-principle demonstration, there are certainly nonidealities in our system. For instance, the transmon decoherence time (few $\mu$s) in our hardware is noticeably lower compared with fixed-frequency transmons[29,43] as well as planar devices specifically optimised for flux noise insensitivity[44,45]. We expect significant improvements in future iterations that incorporate the known mitigating strategies such as alternative circuit designs[44,45] and further optimisation of the hose strength to reduce SQUID loop sizes. We believe that this work conclusively demonstrates the appeal and viability of our strategy, and will prompt future exploration towards more robust and tunable bosonic cQED hardware implementations.

Our results also hint at a much wider range of applications using bosonic cQED architectures equipped with fast-flux tunability. For example, resonant control can be expanded to synthesize arbitrary qudit states[19] and enact individual transitions in the cavity spectrum[23]. Dynamically decoupling the transmon can mitigate the cross-talk in multi-cavity quantum processor architectures. Controlling the idle cavity evolution can be used to optimise tomography protocols, which commonly suffer from coherent errors due to spurious transmon coupling during the measurement[46]. The ability to put different light-matter dynamics in juxtaposition also opens up further possibilities for trotterising Hamiltonian for quantum simulation and computation[47,48]. We also look forward to the application of fast flux for the modulation of the coupling strength $g(t)$ with the high-Q cavity[49–51], for the protection of the transmon from flux noise by leveraging dynamical sweet spots[52–54], and for the exploration of interesting physical models such as the anisotropic Rabi model[55,56] and single-atom lasers[57].

Achieving an architecture capable of tuning the coherent interactions of light and matter on the fly has been a longstanding milestone across many quantum hardware platforms. Our implementation in bosonic cQED provides a highly promising strategy towards addressing this considerable engineering challenge of balancing good cavity lifetimes with fast on-demand flux tunability. Our results bring a

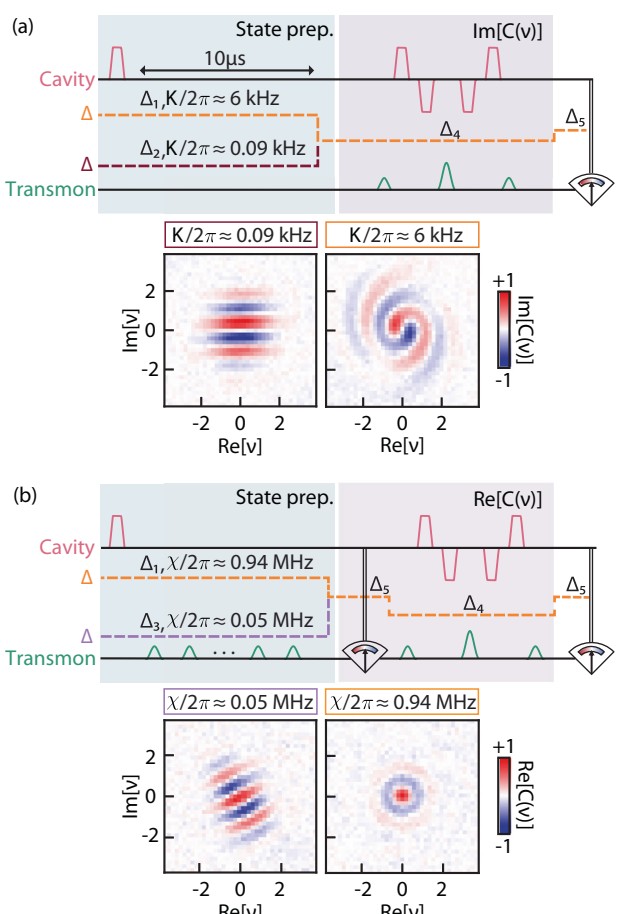

**Fig. 4 | Suppression of undesired cavity dynamics. a** Mitigation of cavity self-Kerr. The cavity is prepared at coherent state $|\alpha = 2.5\rangle$ and allowed to evolve for 10 μs at $\Delta_1/2\pi = 63$ MHz ($K_1/2\pi = 6$ kHz) and $\Delta_2/2\pi = 196$ MHz ($K_2/2\pi = 0.09$ kHz). The imaginary part of the characteristic function $C(v)$ is measured at the end, showing that the state is virtually undisturbed with $K_2$, but significantly distorted during evolution with $K_1$. **b** Decoupling between transmon and cavity by turning off the dispersive interaction. The protocol is similar to the previous experiment, but the transmon is excited by a sequence of 10 $\frac{\pi}{2}$-pulses, each followed by an interval of 400 ns. The real part of the characteristic function of the final state shows that the transmon excitation and decoherence cause the complete loss of phase information from the cavity state at $\chi/2\pi = 0.94$ MHz. In contrast, the coherent state still preserves its phase after evolution at $\chi/2\pi \approx 0.05$ MHz ($\Delta_3/2\pi = 596$ MHz). In both experiments, the tomography and the transmon readout are performed at detunings of $\Delta_4/2\pi = 146$ MHz and $\Delta_5/2\pi = 101$ MHz.

set of powerful functionalities into the repertoire of high-quality superconducting cavities that will stimulate the creation of future protocols for quantum information processing. Furthermore, with this newfound feature, the bosonic cQED platform can now unlock and more effectively harness the vast potential of the full range of dynamics in light-matter interaction.

## Data availability
Source data are provided with this paper. The data and post-processing scripts are available on GitHub (https://github.com/Qcrew/Valadares-NC-ODT). Source data are provided with this paper.

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

## Acknowledgements

We thank Prof. Gerhard Kirchmair, Dr. Stefan Oleschko, and Dr. Ian Yang for their valuable input on the initial development and construction of the magnetic hose. We thank Dr. Guangqiang Liu and Prof. Michel Devoret for providing the quantum amplifier. We thank Adrian Copetudo for the assistance in the chip fabrication. Y. Y. G. acknowledges the support by the Ministry of Education, Singapore, under grant ID T2EP50222-0017 and the National Research Foundation, Singapore, under grant ID NRFF12-2020-0063.

## Author contributions

F.V. conceived and planned the experiment. F.V., N.-N. H., and K.T.N.C. performed the experiment. F.V., N.-N. H., and K.T.N.C. analysed the data. L.K. contributed to the initial device prototyping. A.D. contributed to the analysis and discussion of the results. W.C. and P.S. fabricated the transmon devices. F.V. wrote the manuscript with assistance from N.-N.H., K.T.N.C., A.D., and Y.Y.G. All work was carried out under the supervision of Y.Y.G.

## Competing interests
