## [Peer Review File · Nature Communications]

REVIEWER COMMENTS

Reviewer #1 (Remarks to the Author):

In this work, "On-demand transposition across light-matter interaction regimes in bosonic cQED," the authors presented an experimental implementation showcasing rapid switching across diverse interaction regimes in a bosonic circuit quantum electrodynamics (cQED) system. Their system involves coupling a high-quality-factor superconducting cavity to a non-linear circuit element, the transmon. The achieved tunability between different interaction regimes is enabled by integrating a magnetic hose within a standard bosonic cQED device, facilitating the adjustment of the transmon frequency through magnetic flux manipulation.

The authors demonstrated the practicality of their approach through various applications: (1) fast creation of cavity Fock states using resonant interaction, (2) interchanging tomography techniques at weak and strong dispersive interaction regimes, (3) suppression of unwanted cavity self-Kerr dynamics, and (4) decoupling between transmon and cavity by turning off the dispersive interaction. The fast flux tunability of their system provides a versatile tool for probing light-matter interactions and enhancing quantum information processing.

In conclusion, the results presented in this work are notable, offering potential interest to readers and suggesting avenues for future innovations in the field. I recommend the publication of this article in Nature Communications, pending the successful addressing of the comments outlined below.

Major comments:

1. To enhance the persuasiveness of the demonstrated applications, I suggest that the authors refine the comparisons with the intended outcome states. Specifically, it would be beneficial for the authors to incorporate a quantitative assessment of the performance of their Fock state creation and the suppression of undesired cavity dynamics. This can be achieved by introducing the fidelity for the states in Fig. 3(c) and Fig. 4(a)(b).

Additionally, I highly recommend the inclusion of a tomograph of the coherent state immediately following its preparation, prior to its evolution, in Fig. 4(a)(b).

2. The switching between various interaction regimes in the system is achieved by tuning the transmon frequency. However, it prompts consideration of how this method compares to alternative approaches, such as adjusting the coupling strength, possibly through a tunable coupler [Ref1]. A

more comprehensive exploration and comparison of these different schemes would significantly enhance the understanding and insights from this study.

[Ref1] Sung et al. "Realization of high-fidelity CZ and ZZ-free iSWAP gates with a tunable coupler." *Physical Review X* 11.2, 021058 (2021).

Minor comments:

1. I suggest the authors clarify technical terms and include a broader range of references to enhance accessibility to a wider audience.

A. P.1, left column, "Many physical platforms originate from the interactions between light and matter [1-4]": Missing circuit quantum acoustodynamics systems in the reference, for example [Ref2] Manenti et al. "Circuit quantum acoustodynamics with surface acoustic waves." *Nature Communications* 8.1, 975 (2017)

B. P.2, left column, "by tuning the cavity-transmon interaction to the strong and weak dispersive regimes, respectively": Adding discussions on dispersive couplings and different coupling regimes will be helpful, for example: [Ref3] Schuster et al. "Resolving photon number states in a superconducting circuit." *Nature* 445.7127, 515 (2007). [Ref4] Blais et al. "Quantum-information processing with circuit quantum electrodynamics." *Physical Review A* 75.3, 032329 (2007).

C. Please consider providing the complete form of the acronyms SQUID (p.1 right column), FPGA (p.3 right column), and RF (p.3 right column).

D. P.3, right column, "These distortions mainly come from the inductance of the coil and the RF port of the bias tee, and can be reverted using the method described in Ref. [16, 29]": Please consider briefly describing the method in the main text.

E. P.4, left column, "the creation of non-Gaussian cavity states through vacuum Rabi oscillations (Fig. 3a)": Please bridge the connection between "non-Gaussian states" and "Fock states".

F. P.4, right column, "motivating the search for ways to control and mitigate nonlinearities using drives or alternative circuits [36–38]": Some relevant references on Kerr nonlinearity cancellation are missing, for example [Ref5] Wang et al. "Photon-number-dependent Hamiltonian engineering for cavities." *Physical Review Applied* 15.4, 044026 (2021). [Ref6] Zhang et al. "Drive-induced nonlinearities of cavity modes coupled to a transmon ancilla." *Physical Review A* 105.2, 022423 (2022).

G. A review of the control of bosonic cQED may be included in the manuscript: [Ref7] Ma et al. "Quantum control of bosonic modes with superconducting circuits." *Science Bulletin* 66.17, 1789-1805 (2021).

2. P.4, left column, "State $|1\rangle$ is prepared in resonance after an interaction time $\tau_{\text{int}} \approx 30$ ns, while $|2\rangle$ is prepared by repeating the same protocol twice with total $\tau_{\text{int}} \approx 50$ ns": Why does repeating the protocol not amount to doubling the interaction time?

Reviewer #2 (Remarks to the Author):

This paper reports realization of coupling between a 3D cavity and a tunable transmon preserving a long coherence time of the cavity. A key device for this achievement is a carefully designed magnetic hose, proposed and demonstrated in References 27 and 28. As demonstration of the high coherence and tunability of their system, they performed the Fock state generation and their tomography, and confirmed the clean time evolution by increasing the cavity-transmon detuning.

Since tuning of transmon frequency by magnetic flux is a well-established technique in circuit QED and the demonstrated phenomena have already been achieved in previous works, the novelty of this work is limited to technical points. I feel that this paper deserves publication in some form but does not meet the acceptance criteria for this journal.

Comments

(1) In the lower half of Figure 1, the meanings of three cavity-atom systems are unclear.

(2) The authors wrote "transposition across several distinct interaction regimes" and "the many regimes of light-matter interactions". I could find only two regimes (resonant and decoupled). What are the other regimes?

(3) There are only few papers which refer to a transmon-cavity coupled system as a bosonic circuit QED. Therefore, the statement that "this architecture, known as bosonic cQED" needs revision. Furthermore, since one would readily associate the word "bosonic cQED" with a system without nonlinearity, this is quite a misleading name for the transmon-cavity system.

(4) In equation (1), ω_t and H should be replaced with $\omega_t(t)$ and $H(t)$.

Point-by-point Response to the reviewers

We thank the reviewers for their overall positive assessment of our work. In the following, we address their questions and comments point-by-point.

Reviewer 1

In this work, “On-demand transposition across light-matter interaction regimes in bosonic cQED,” the authors presented an experimental implementation showcasing rapid switching across diverse interaction regimes in a bosonic circuit quantum electrodynamics (cQED) system. Their system involves coupling a high-quality-factor superconducting cavity to a non-linear circuit element, the transmon. The achieved tunability between different interaction regimes is enabled by integrating a magnetic hose within a standard bosonic cQED device, facilitating the adjustment of the transmon frequency through magnetic flux manipulation.

The authors demonstrated the practicality of their approach through various applications: (1) fast creation of cavity Fock states using resonant interaction, (2) interchanging tomography techniques at weak and strong dispersive interaction regimes, (3) suppression of unwanted cavity self-Kerr dynamics, and (4) decoupling between transmon and cavity by turning off the dispersive interaction. The fast flux tunability of their system provides a versatile tool for probing light-matter interactions and enhancing quantum information processing.

In conclusion, the results presented in this work are notable, offering potential interest to readers and suggesting avenues for future innovations in the field. I recommend the publication of this article in Nature Communications, pending the successful addressing of the comments outlined below.

We thank the reviewer for the detailed read of our manuscript and the firm endorsement for publication in Nature Communications. We also appreciate very much the salient questions raised in the the review and will address them in detail below.

Reviewer Point P 1.1 — To enhance the persuasiveness of the demonstrated applications, I suggest that the authors refine the comparisons with the intended outcome states. Specifically, it would be beneficial for the authors to incorporate a quantitative assessment of the performance of their Fock state creation and the suppression of undesired cavity dynamics. This can be achieved by introducing the fidelity for the states in Fig. 3(c) and Fig. 4(a)(b). Additionally, I highly recommend the inclusion of a tomograph of the coherent state immediately following its preparation, prior to its evolution, in Fig. 4(a)(b).

Reply: We appreciate the reviewer’s suggestions. We compute the fidelity of the states of interest by directly calculating the overlap integral of the corresponding ideal Wigner function and characteristic function over the phase space. For a state with a characteristic function denoted as $C_{\text{exp}}(\nu)$, the overlap with the ideal target state $C_{\text{ideal}}(\nu)$ is calculated using the equation:

$$\mathcal{F}_{\text{int}} = \frac{1}{\pi} \int C_{\text{ideal}}(\nu) C_{\text{exp}}^*(\nu) d^2\nu. \quad (1)$$

Similarly, for a state with a Wigner function, $W_{\text{exp}}(\beta)$, we compute the overlap with the ideal target state $W_{\text{ideal}}(\beta)$ using the following equation:

$$\mathcal{F}_{\text{int}} = \frac{1}{\pi} \int W_{\text{ideal}}(\beta) W_{\text{exp}}(\beta) d^2\beta. \quad (2)$$

Since our data is discrete, we used a summation instead of an integral to compute the fidelities. The calculated fidelities for the Wigner (characteristic) functions of the Fock states 1 and 2 are 91.1 (82.8) (%) and 66.6 (62.8) (%), respectively.

The reduced fidelity is attributed to the 80 ns qubit π -pulse being comparable in length (a factor of 0.2) to the qubit coherence time T_2 (400 ns). The comparison between the measured data and simulation (Fig. 10 in the Supplemental materials) shows that the prepared states exhibit all the key features in phase space. A full simulation that takes into account the finite pulse duration and short T_2 time of the qubit also indicates a consistent drop in the state creation fidelity.

We emphasize that this limited fidelity is not a limitation of our strategy of creating non-Gaussian states using resonant coupling enabled by the fast flux, but a rather device-specific experimental woe. The short T_2 in this particular transmon device is most likely due to two experimental factors: 1. the large SQUID loop and thus, increased susceptibility to ambient magnetic noise, which can be improved by adopting existing techniques such as making the SQUID asymmetric; and 2. device fabrication, as we have noticed that many standard transmons fabricated during the same period in the group also exhibits low T_2 . Therefore, we are confident that the overall technique is capable of achieving fast and high-fidelity state creation with some improvements in our device design and fabrication.

We have expanded the analysis and discussion of the fidelities in the supplemental materials to include these new details.

As for the coherent state after preparation, we did not retain the fine-grained 2-dimensional tomography data of the state right after being prepared since it would be seemingly redundant compared to the left graph of Fig. 4a. However, we did perform thorough control experiments to ensure both the quality of our tomography process as well as the state preparation.

We calibrate the tomography by measuring the cross-section of the characteristic function of vacuum: the standard variation of the real part (see Figure below) gives the scale of the reciprocal-space displacement, the amplitude of the real part gives the maximum contrast of the tomography, while the imaginary part can be calibrated into showing a flat profile by adjusting the phase of the second $\frac{\pi}{2}$ -pulse of the protocol.

1D cut of the real (blue) and imaginary (red) parts of the characteristic function of vacuum used for tomography calibration.

To illustrate the quality of arbitrary coherent state creation, we show here the tomography data of a target coherent state right after preparation in a control experiment to ensure the quality of the initial state:

Measured data of the real part of the characteristic function of a coherent state. The fit indicates that a coherent state of $\alpha = 1.45$ is created, which matches with the target state of $\alpha = 1.5$ within statistical errors.

Reviewer Point P 1.2 — The switching between various interaction regimes in the system is achieved by tuning the transmon frequency. However, it prompts consideration of how this method compares to alternative approaches, such as adjusting the coupling strength, possibly through a tunable coupler [Ref1]. A more comprehensive exploration and comparison of these different schemes would significantly enhance the understanding and insights from this study. [Ref1] Sung et al. “Realization of high-fidelity CZ and ZZ-free iSWAP gates with a tunable coupler.” Physical Review X 11.2, 021058 (2021).

Reply: We agree with the reviewer that including this discussion will enrich and better contextualize the current work. We consider the tunable coupler strategy as a complementary instead of competing effort, since it would also require non-intrusive fast-flux tunability. In addition, as our implementation does not require a dedicated coupler mode, it is more advantageous for multi-mode systems that may face issues of on-chip frequency crowding or cross-talk. In this spirit, we expanded the outlook discussion to include these topics as future work/applications (p.5 right column).

Reviewer Point P 1.3 — I suggest the authors clarify technical terms and include a broader range of references to enhance accessibility to a wider audience.

1. P.1, left column, “Many physical platforms originate from the interactions between light and matter [1-4]”: Missing circuit quantum acoustodynamics systems in the reference, for example [Ref2] Manenti et al. “Circuit quantum acoustodynamics with surface acoustic waves.” Nature Communications 8.1, 975 (2017)
2. P.2, left column, “by tuning the cavity-transmon interaction to the strong and weak dispersive regimes, respectively”: Adding discussions on dispersive couplings and different coupling regimes will be helpful, for example: [Ref3] Schuster et al. “Resolving photon number states in a superconducting circuit.” Nature 445.7127, 515 (2007). [Ref4] Blais et al. “Quantum-information processing with circuit quantum electrodynamics.” Physical Review A 75.3, 032329 (2007).

3. Please consider providing the complete form of the acronyms SQUID (p.1 right column), FPGA (p.3 right column), and RF (p.3 right column).
4. P.3, right column, “These distortions mainly come from the inductance of the coil and the RF port of the bias tee, and can be reverted using the method described in Ref. [16, 29]”: Please consider briefly describing the method in the main text.
5. P.4, left column, “the creation of non-Gaussian cavity states through vacuum Rabi oscillations (Fig. 3a)”: Please bridge the connection between “non-Gaussian states” and “Fock states”.
6. P.4, right column, “motivating the search for ways to control and mitigate nonlinearities using drives or alternative circuits [36–38]”: Some relevant references on Kerr nonlinearity cancellation are missing, for example [Ref5] Wang et al. “Photon-number-dependent Hamiltonian engineering for cavities.” *Physical Review Applied* 15.4, 044026 (2021). [Ref6] Zhang et al. “Drive-induced nonlinearities of cavity modes coupled to a transmon ancilla.” *Physical Review A* 105.2, 022423 (2022).
7. A review of the control of bosonic cQED may be included in the manuscript: [Ref7] Ma et al. “Quantum control of bosonic modes with superconducting circuits.” *Science Bulletin* 66.17, 1789-1805 (2021).

Reply: We appreciate and agree with the reviewer’s suggestions. We have implemented them accordingly. To explain some of the changes:

- Points 1,3 and 6 have been implemented as suggested
- Point 2: we have added a general introduction to the different interaction regimes and included the suggested references to the caption of Fig. 1. We have also added a statement to P.2, left column to provide additional details.
- Point 4: in this excerpt we were referring to the method that would be described in the rest of the paragraph. The rephrasing should make this connection clearer.
- Points 5 and 7: we realized the expression “non-Gaussian states” was being overused. We replaced most instances for “Fock states”, and then used the publication by Ma *et al.* to link Fock states to the non-Gaussian resources required for universal quantum computing (p.4 left column).

Reviewer Point P 1.4 — P.4, left column, “State $|1\rangle$ is prepared in resonance after an interaction time $\tau_{\text{int}} \approx 30$ ns, while $|2\rangle$ is prepared by repeating the same protocol twice with total $\tau_{\text{int}} \approx 50$ ns”: Why does repeating the protocol not amount to doubling the interaction time?

Reply: We thank the reviewer for pointing this out and we agree that the previous text did not provide sufficient details for the reader. We added a sentence explaining that the Rabi frequency of the second transition increases by a factor of $\sqrt{2}$, which explains why the duration of the protocol doesn’t scale linearly. Additionally, we also cited Ref. [1] where the reader will find detailed discussions about the Jaynes-Cummings energy spectrum.

Reviewer 2

This paper reports realization of coupling between a 3D cavity and a tunable transmon preserving a long coherence time of the cavity. A key device for this achievement is a carefully designed magnetic hose, proposed and demonstrated in References 27 and 28. As demonstration of the high coherence and tunability of their system, they performed the Fock state generation and their tomography, and confirmed the clean time evolution by increasing the cavity-transmon detuning.

Since tuning of transmon frequency by magnetic flux is a well-established technique in circuit QED and the demonstrated phenomena have already been achieved in previous works, the novelty of this work is limited to technical points. I feel that this paper deserves publication in some form but does not meet the acceptance criteria for this journal.

We thank the reviewer for taking the time to assess our work and affirming the main achievements we reported here. However, we do not agree with the comment that the novelty of this work is limited to technical points and we elaborate our counter-arguments below.

First, the tuning of transmon frequency by magnetic flux in a cQED system with high-Q cavities is actually a highly desirable feature and a long-standing challenge faced by the community. In fact, earlier attempts (Chapter 9 of Ref. [2]) have conclusively indicated that naive incorporation flux-tunability dramatically reduces both the cavity and transmon lifetimes. Thus, the ability to introduce both DC and AC flux-tunability while ensuring long coherence times of the cavity mode shown in this work is a significant achievement and a valuable addition to the capabilities of cQED devices that will open up new possibilities for quantum simulation, control, and measurements.

A second point mentioned by the reviewer refers to the phenomena we demonstrated in this work, namely, the creation of non-Gaussian states through fast, resonant control and the tuning of transmon-cavity coupling regimes on-demand to suppress unwanted evolutions on the bosonic state. They are highly valuable for continuous-variable quantum systems and have not been demonstrated in any prior experiments of comparable capabilities. For instance, using resonant control to perform state preparation on a continuous-variable system was shown only on planar devices with very low coherence times, which are not suitable candidates for encoding and storing information. Our ability to do so on high-Q cavities thus affords a whole new suite of new opportunities for realising robust information encoding and storage. Moreover, the suppression of undesired evolution by switching the interaction between the transmon and bosonic mode on the fly is a sourly-missed ingredient for quantum error correction. As discussed in Refs. [3], the spurious, always-on coupling is one of the main limitations on the logical lifetime of a cat qubit encoded in a superconducting cavity, and a similar susceptibility applies to other single-mode bosonic codes [4]. With the phenomema we showcased in the study, this issue can now be addressed effectively without sacrificing critical requirements such as the intrinsic cavity coherence times and robust parity mapping of bosonic QEC schemes. We are not aware, to the best of our knowledge, of any prior implementations of such capability. Finally, by allowing

the coupling to be adjusted on-demand, we also enabled access to quantitatively different control and measurement techniques in one device, e.g. creating a quantum state using resonant control and performing tomography using the characteristic function, which requires very low non-linear coupling. As far as we know, there are no existing implementations that offer such a possibility.

Therefore, we argue that the demonstrations shown in our work are both novel and impactful, as they address some of the most critical challenges in continuous-variable based quantum information processing. We firmly believe that our work very adequately meets the standard of Nature Communications. We are appreciative of the concern raised by the reviewer and have improved the clarity of our impact statements in the manuscript to ensure that these key points are more effectively articulated.

Below, we address the remaining questions from the reviewer point-by-point.

Reviewer Point P 2.1 — In the lower half of Figure 1, the meanings of three cavity-atom systems are unclear.

Reply: We thank the reviewer for the feedback on the clarity. To address this point and the next, we have rewritten the caption of Fig. 1 to include the following

- We classify each regime according to g/Δ to better match the figure.
- The decoupled regime is mentioned first and the resonant regime second to better match the reading order of the figure.
- Extended explanation of the dispersive regime.

Reviewer Point P 2.2 — The authors wrote “transposition across several distinct interaction regimes” and “the many regimes of light-matter interactions”. I could find only two regimes (resonant and decoupled). What are the other regimes?

Reply: The reviewer correctly points out that we did not formally define the strong and weak dispersive coupling regimes. While these regimes might appear to be the same “decoupled” regime on the surface, they result in qualitatively different dynamics that must be carefully taken into account experimentally. Each of these regimes also enables access to dramatically different control and measurement techniques. For instance, in the strong dispersive regime it is possible to enact pulses on the transmon that are selective of the cavity state, and map the parity of the cavity onto the transmon, allowing Wigner tomography. Although we cannot use these resources in the weak dispersive regime, it is possible instead to implement universal control based on echoed conditional displacement gates and characterize the state with Characteristic function tomography [5]. To emphasize this point, we have added a clarifying statement in P.2, left column. We have also expanded the introduction of the different regimes in the caption of Fig. 1 and included the following references for further reading: [6, 7].

Reviewer Point P 2.3 — There are only few papers which refer to a transmon-cavity coupled system as a bosonic circuit QED. Therefore, the statement that “this architecture, known as bosonic cQED” needs revision. Furthermore, since one would readily associate the word “bosonic cQED” with a system without nonlinearity, this is quite a misleading name for the transmon-cavity system.

Reply: The reviewer has correctly highlighted that “bosonic cQED” is a relatively new term that might be unfamiliar to some readers. To address this, we have rewritten the sentence where bosonic cQED is introduced to give a clearer definition. Furthermore, we moved references [8,9] to the line in question so readers will be able to clarify any other doubts.

Overall, we understand that lacking the context of the specific challenges of bosonic cQED one might take the contributions of this paper as merely technical. But in fact, our achievements present the first solution to a long-standing problem and offer an entirely new axis of control for continuous-variable experiments in superconducting platforms. We highlight that many of the cited papers either propose algorithms that require such a solution to be useful for any information-processing task [10–19]; or have already been implemented in circuit QED but cannot be used as routines in continuous-variable experiments because they are limited by the cavity lifetime [20–24].

As an example, we quote the conclusion of Ref. [15]:

These first proposals [...] provide broad and valuable tools for hybrid continuous-discrete variable quantum information processing and simulations of nonlinear bosonic systems. However, implementing this method needs further experimental development, including switching a single qubit between resonant and dispersive coupling with multiple cavity modes, a capability never tested in current superconducting circuit QED and ion trap systems limited to fixed interaction regimes.

Similarly, Ref. [25], Terhal et al. discusses the advantages of tunable cavity-qubit coupling for the preparation of high- \bar{n} GKP states:

In some physical set-ups the dispersive cavity-qubit coupling (both storage and read-out cavity) is not tunable and is thus always ‘on’. Such a setup is non-ideal in various ways. When the dispersive coupling is always on, it means that one should prepare the code states in a rotating frame (not the lab frame) which depends on the qubit state. [...] Another disadvantage of using a non-tunable χ is that the accuracy of single qubit rotations depends on the number of photons in the cavity. [...] In [16] it was argued that the unwanted entangling of qubit and cavity due to single qubit rotations is a leading source of inaccuracies when one goes to higher photon numbers. [...] A third disadvantage of a non-tunable χ is the the cavity Kerr nonlinearity which is present in Eq. (25) due to the linear coupling between the LC oscillators: the “cavity mode” is in fact a “dressed” cavity mode which sees the Josephson nonlinearity. If χ is turned to a small value, then this Kerr nonlinearity will be correspondingly small.

From these statements, it is evident that the capabilities we demonstrated in this work are of great value to the general quantum information processing community. Thus, we are confident about the broad scientific significance and impact of our results.

Reviewer Point P 2.4 — (4) In equation (1), ω_t and H should be replaced with $\omega_t(t)$ and $H(t)$.

Reply: We thank the reviewer for the correction and have implemented the indicated changes.

References

- [1] Alexandre Blais, Arne L. Grimsmo, S.M. Girvin, and Andreas Wallraff. Circuit quantum electrodynamics. *Reviews of Modern Physics*, 93(2):025005, May 2021.

- [2] Matthew David Reed. *Entanglement and Quantum Error Correction with Superconducting Qubits*. PhD, Yale University, May 2013.
- [3] Nissim Ofek, Andrei Petrenko, Reinier Heeres, Philip Reinhold, Zaki Leghtas, Brian Vlastakis, Yehan Liu, Luigi Frunzio, S. M. Girvin, L. Jiang, Mazhar Mirrahimi, M. H. Devoret, and R. J. Schoelkopf. Extending the lifetime of a quantum bit with error correction in superconducting circuits. *Nature*, 536(7617):441–445, August 2016.
- [4] Victor V. Albert, Kyungjoo Noh, Kasper Duivenvoorden, Dylan J. Young, R. T. Brierley, Philip Reinhold, Christophe Vuillot, Linshu Li, Chao Shen, S. M. Girvin, Barbara M. Terhal, and Liang Jiang. Performance and structure of single-mode bosonic codes. *Phys. Rev. A*, 97:032346, Mar 2018.
- [5] Alec Eickbusch, Volodymyr Sivak, Andy Z Ding, Salvatore S Elder, Shantanu R Jha, Jayameenakshi Venkatraman, Baptiste Royer, Steven M Girvin, Robert J Schoelkopf, and Michel H Devoret. Fast universal control of an oscillator with weak dispersive coupling to a qubit. *Nature Physics*, 18(12):1464–1469, 2022.
- [6] Alexandre Blais, Jay Gambetta, Andreas Wallraff, David I Schuster, Steven M Girvin, Michel H Devoret, and Robert J Schoelkopf. Quantum-information processing with circuit quantum electrodynamics. *Physical Review A*, 75(3):032329, 2007.
- [7] DI Schuster, Andrew Addison Houck, JA Schreier, A Wallraff, JM Gambetta, A Blais, L Frunzio, J Majer, B Johnson, MH Devoret, et al. Resolving photon number states in a superconducting circuit. *Nature*, 445(7127):515–518, 2007.
- [8] Atharv Joshi, Kyungjoo Noh, and Yvonne Y Gao. Quantum information processing with bosonic qubits in circuit QED. *Quantum Science and Technology*, 6(3):033001, July 2021.
- [9] Adrian Copetudo, Clara Yun Fontaine, Fernando Valadares, and Yvonne Y. Gao. Shaping photons: Quantum information processing with bosonic cQED. *Applied Physics Letters*, 124(8):080502, 02 2024.
- [10] Roshan Sharma and Frederick W. Strauch. Quantum state synthesis of superconducting resonators. *Physical Review A*, 93(1):012342, January 2016.
- [11] Frederick W. Strauch, Kurt Jacobs, and Raymond W. Simmonds. Arbitrary Control of Entanglement between two Superconducting Resonators. *Physical Review Letters*, 105(5):050501, July 2010.
- [12] Frederick W. Strauch. All-Resonant Control of Superconducting Resonators. *Physical Review Letters*, 109(21):210501, November 2012.
- [13] Takaaki Aoki, Taro Kanao, Hayato Goto, Shiro Kawabata, and Shumpei Masuda. Control of the $\$ZZ\$$ coupling between Kerr-cat qubits via transmon couplers, March 2023. arXiv:2303.16622 [quant-ph].
- [14] Qi-Kai He and Duan-Lu Zhou. Tunable coupling between a superconducting resonator and an artificial atom. *The European Physical Journal D*, 73(5):96, May 2019.

- [15] Kimin Park, Petr Marek, and Radim Filip. Efficient quantum simulation of nonlinear interactions using snap and rabi gates. *Quantum Science and Technology*, 9(2):025004, January 2024.
- [16] Youngkyu Sung, Leon Ding, Jochen Braumüller, Antti Vepsäläinen, Bharath Kannan, Morten Kjaergaard, Ami Greene, Gabriel O. Samach, Chris McNally, David Kim, Alexander Melville, Bethany M. Niedzielski, Mollie E. Schwartz, Jonilyn L. Yoder, Terry P. Orlando, Simon Gustavsson, and William D. Oliver. Realization of high-fidelity cz and zz-free iswap gates with a tunable coupler. *Phys. Rev. X*, 11:021058, Jun 2021.
- [17] Gangcheng Wang, Ruoqi Xiao, H. Z. Shen, Chunfang Sun, and Kang Xue. Simulating anisotropic quantum rabi model via frequency modulation. *Scientific Reports*, 9(1), March 2019.
- [18] Yimin Wang, Wen-Long You, Maoxin Liu, Yu-Li Dong, Hong-Gang Luo, G Romero, and J Q You. Quantum criticality and state engineering in the simulated anisotropic quantum rabi model. *New Journal of Physics*, 20(5):053061, May 2018.
- [19] Stephan André, Pei-Qing Jin, Valentina Brosco, Jared H. Cole, Alessandro Romito, Alexander Shnirman, and Gerd Schön. Single-qubit lasing in the strong-coupling regime. *Physical Review A*, 82(5), November 2010.
- [20] Max Hofheinz, H. Wang, M. Ansmann, Radoslaw C. Bialczak, Erik Lucero, M. Neeley, A. D. O’Connell, D. Sank, J. Wenner, John M. Martinis, and A. N. Cleland. Synthesizing arbitrary quantum states in a superconducting resonator. *Nature*, 459(7246):546–549, May 2009. Number: 7246 Publisher: Nature Publishing Group.
- [21] A. Mezzacapo, U. Las Heras, J. S. Pedernales, L. DiCarlo, E. Solano, and L. Lamata. Digital Quantum Rabi and Dicke Models in Superconducting Circuits. *Scientific Reports*, 4(1):7482, December 2014. Number: 1 Publisher: Nature Publishing Group.
- [22] R. K. Naik, N. Leung, S. Chakram, Peter Groszkowski, Y. Lu, N. Earnest, D. C. McKay, Jens Koch, and D. I. Schuster. Random access quantum information processors using multimode circuit quantum electrodynamics. *Nature Communications*, 8(1), December 2017.
- [23] Yulin Wu, Li-Ping Yang, Ming Gong, Yarui Zheng, Hui Deng, Zhiguang Yan, Yanjun Zhao, Keqiang Huang, Anthony D. Castellano, William J. Munro, Kae Nemoto, Dong-Ning Zheng, C. P. Sun, Yu-xi Liu, Xiaobo Zhu, and Li Lu. An efficient and compact switch for quantum circuits. *npj Quantum Information*, 4(1), October 2018.
- [24] Sabrina S. Hong, Alexander T. Papageorge, Prasahnt Sivarajah, Genya Crossman, Nicolas Didier, Anthony M. Polloreno, Eyob A. Sete, Stefan W. Turkowski, Marcus P. da Silva, and Blake R. Johnson. Demonstration of a parametrically activated entangling gate protected from flux noise. *Physical Review A*, 101(1), January 2020.
- [25] B. M. Terhal and D. Weigand. Encoding a qubit into a cavity mode in circuit QED using phase estimation. *Physical Review A*, 93(1):012315, January 2016.

REVIEWERS' COMMENTS

Reviewer #1 (Remarks to the Author):

The authors have adequately addressed my comments and revised the manuscript accordingly; therefore, I recommend the manuscript for publication.

Reviewer #2 (Remarks to the Author):

By reading the response by the authors, I realized the significance of this work in the context of implementation of bosonic qubits. I withdraw my previous judgement and recommend publication of this paper, after revisions as follows.

Comments:

I think the authors can show the "trace" of the authors' system in the phase diagram of cQED [Fig. 1 of Nature 445, 515 (2007)] and I hope the authors to add it on the occasion of further revision. This figure would support the title of this paper "transposition across light-matter interaction regimes".

In the previous reviewing, I commented that the meanings of three cavity-atom systems are unclear. In the revised manuscript, I still feel so unfortunately. What do the wavy lines with different colors mean in the middle figure? Does the rightmost figure represent the vacuum Rabi oscillation? This figure does not look like that.

Point-by-point Response to the reviewers

We thank the reviewers for their overall positive assessment of our work. In the following, we address their questions and comments point-by-point.

Reviewer 1

The authors have adequately addressed my comments and revised the manuscript accordingly; therefore, I recommend the manuscript for publication.

We thank the reviewer for the careful and fair assessment of our manuscript.

Reviewer 2

By reading the response by the authors, I realized the significance of this work in the context of implementation of bosonic qubits. I withdraw my previous judgement and recommend publication of this paper, after revisions as follows.

We thank the reviewer for the reconsideration of the manuscript and suggestions for further improvement. Below we address both points mentioned by the reviewer.

Reviewer Point P 2.1 — I think the authors can show the "trace" of the authors' system in the phase diagram of cQED [Fig. 1 of Nature 445, 515 (2007)] and I hope the authors to add it on the occasion of further revision. This figure would support the title of this paper "transposition across light-matter interaction regimes".

Reply: We thank the reviewer for the suggestion. We have included a qualitative depiction of the diagram and a trace representing the regimes accessed by our system in Supplementary Figure 1.

Reviewer Point P 2.2 — In the previous reviewing, I commented that the meanings of three cavity-atom systems are unclear. In the revised manuscript, I still feel so unfortunately. What do the wavy lines with different colors mean in the middle figure? Does the rightmost figure represent the vacuum Rabi oscillation? This figure does not look like that.

Reply: We appreciate the reviewer's feedback on the clarity of Figure 1. To address these concerns, we have made the following revisions:

- We included labels to each cavity-atom system to avoid ambiguity. We also mark the position where each regime falls in the g/Δ axis.
- The atom is represented as a two-level atomic system with distinct ground and excited states. This way, the third figure indicates more clearly that there can be resonant exchange of energy through vacuum Rabi oscillations.

Regarding the middle cavity-atom system, the two different colors show the energy shift the atom can impart on the cavity depending on its state while on the dispersive regime. With the labeling of each regime, we expect the meaning to be clearer.